# The noise level in linear regression with dependent data

**Ingvar Ziemann**
University of Pennsylvania

**Stephen Tu**
Google Research

**George J. Pappas**
University of Pennsylvania

**Nikolai Matni**
University of Pennsylvania

## Abstract

We derive upper bounds for random design linear regression with dependent ($\beta$-mixing) data absent any realizability assumptions. In contrast to the strictly realizable martingale noise regime, no sharp *instance-optimal* non-asymptotics are available in the literature. Up to constant factors, our analysis correctly recovers the variance term predicted by the Central Limit Theorem—the noise level of the problem—and thus exhibits graceful degradation as we introduce misspecification. Past a burn-in, our result is sharp in the moderate deviations regime, and in particular does not inflate the leading order term by mixing time factors.

## 1 Introduction

Ordinary least squares (OLS) regression from a finite sample is one of the most ubiquitous and widely used technique in machine learning. When faced with independent data, there are now sharp tools available to analyze its success optimally under relatively general assumptions. Indeed, a non-asymptotic theory matching the classical asymptotically optimal understanding from statistics [van der Vaart, 2000] has been developed over the last decade [Hsu et al., 2012, Oliveira, 2016, Mourtada, 2022]. However, once we relax the independence assumption and move toward data that exhibits correlations, the situation is much less well-understood—even for a problem as seemingly simple as linear regression. While sharp asymptotics are available through various limit theorems, there are no general results matching these in the finite sample regime.

In this paper, we study the instance-specific performance of ordinary least squares in a setting with dependent data—and in contrast to much contemporary work on the theme—without imposing realizability.[1] If in addition to a realizability assumption the noise forms a martingale difference sequence, it is now well-known that martingale methods can be used to demonstrate that dependent linear regression is no harder than its independent counterpart [Simchowitz et al., 2018]. Furthermore, as long as one maintains such an assumption on the noise, a similar observation even holds true for generalized linear and bilinear models [Kowshik et al., 2021, Sattar et al., 2022], and regression with square loss more generally [Ziemann and Tu, 2022].

However, barring any such strong realizability assumption, martingale methods are no longer directly available, and neither are there any sharp non-asymptotics in the learning theory literature. Absent martingale techniques, a natural approach is to use the blocking technique [Bernstein, 1927, Yu, 1994] to port concentration inequalities valid for independent data to the dependent setting. However, since blocking effectively reduces the sample size by a factor of the degree of dependence of the data, a judicious application is necessary in order to recover the correct noise level of the problem—the level predicted by the Central Limit Theorem (CLT).

---

[1]A distribution $\mathsf{P}_{X,Y}$ is (linearly) realizable if the regression function $x \mapsto \mathbf{E}[Y \mid X = x]$ is linear.

37th Conference on Neural Information Processing Systems (NeurIPS 2023).

## 1.1 Contributions

This paper serves to explain how the combination of two simple yet powerful observations sidestep the aforementioned issues with blocking. To better appreciate these observations, we recall that the analysis of random design linear regression decomposes into: (1) controlling the *lower tail* of the empirical covariance matrix; and (2) controlling the interaction between the noise and the covariates.

First, as noted by Mendelson [2014], the dominant contribution to the error rate is due to the interaction of the noise with the covariates via the hypothesis class. In linear regression this interaction term takes the form of a random walk (see (2.6) and (2.8) below). While one must also analyze the lower tail of the empirical covariance matrix, its contribution to the final error tends to be lower order. This is exactly the point: the empirical covariance matrix tends to dominate its population counterpart under very mild assumptions [see e.g. Koltchinskii and Mendelson, 2015, Oliveira, 2016, Simchowitz et al., 2018]. Hence deflating the sample size for this purpose by using dependency is of relatively minor consequence and only amounts to an additional burn-in.

Second, turning to the random walk—the noise-class interaction term—the above issue with blocking can be remedied if one restricts its use to control only the largest scale of deviation. This observation can be traced to the moderate deviations literature, but does not seem to have made its way into the learning theory literature [see e.g. Merlevède et al., 2011]. To explain this idea, let us recall Bernstein's inequality: for $b > 0$, $\delta \in (0, 1)$, and a sequence of $n \in \mathbb{N}$ iid mean zero $b$-bounded scalar random variables $V_{1:n}$,

$$\mathbf{P}\left(\frac{1}{n}\sum_{i=1}^{n} V_i \geq 2\sqrt{\frac{\mathbf{E}V_1^2 \ln(1/\delta)}{n}} + \frac{4b\log(1/\delta)}{3n}\right) \leq \delta. \tag{1.1}$$

In the moderate deviations bandwidth ($\delta \gtrsim \exp(-n\mathbf{E}V_1^2/b^2)$), the leading term of (1.1) is exactly of the expected order, seen from a central limit heuristic: $\sqrt{\frac{\mathbf{E}V_1^2 \ln(1/\delta)}{n}}$. Assume now for sake of argument that $k \in \mathbb{N}$ divides $n$ and set $m = n/k$. Applying (1.1) instead to the $bk$-bounded variables $\bar{V}_{i:m}, \bar{V}_i \triangleq \sum_{j=ik-k-1}^{ik} V_j$ we find instead:

$$\mathbf{P}\left(\frac{1}{n}\sum_{i=1}^{n} V_i \geq 2\sqrt{\frac{k^{-1}\mathbf{E}(\bar{V}_1)^2 \ln(1/\delta)}{n}} + \frac{4bk\log(1/\delta)}{3n}\right) \leq \delta. \tag{1.2}$$

The (normalized) variance of iid random variables tensorizes nicely ($k^{-1}\mathbf{E}(\bar{V}_1)^2 = \mathbf{E}V_1^2$), and so the only difference between (1.1) and (1.2) is that the large deviations term has been inflated by a factor $k$. More generally, however, (1.2) remains valid as long as every $k$ samples are blockwise independent. The leading term of (1.2) already captures the correct variance term in the blockwise independent, one-dimensional and bounded setting.

The above two paragraphs illustrate the core of our argument: by combining the above two observations we can entirely relegate any dependence on mixing to additive burn-in factors. In the sequel, we produce a more general version of this argument. To allow for arbitrary dimensions and handle unbounded processes, we first replace Bernstein's inequality with a corollary to Talagrand's inequality due to Einmahl and Li [2008]. To allow for $\beta$-mixing processes, we replace the blockwise independence assumption with the blocking strategy of Yu [1994]. By combining with control of the lower tail, which as noted above holds under mild assumptions, this leads to our main result Theorem 3.1, captured informally below.

**Informal version of Theorem 3.1.** *Past a mild burn-in, polynomial in relevant problem quantities including the $\beta$-mixing coefficients of the data, and for a fixed failure probability $\delta \in (0, 1)$, OLS with one-dimensional targets and $d_\mathsf{X}$-dimensional covariates enjoys the following excess risk guarantee:*

$$\text{Excess Risk (OLS)} \lesssim \frac{\sigma^2 \left(d_\mathsf{X} + \log(1/\delta)\right)}{n}. \tag{1.3}$$

*Moreover, the term $\sigma^2$ in (1.3) accurately captures the noise level of the problem solely via the relevant second order statistics; it is not inflated by any mixing times.*

The crux of this result is that past a burn-in, the OLS excess risk does not directly depend on mixing times, but only on the relevant second order statistics. Put differently, the effect of slow mixing has

been relegated to a small *additive* term with higher order dependence on $1/n$. This stands in stark contrast to the usual invocation of the blocking technique where the effect of mixing typically enters *multiplicatively*, thereby degrading the rate of convergence uniformly across all sample-sizes and past any burn-in times [see e.g. Steinwart and Christmann, 2009, Kuznetsov and Mohri, 2017, Wong et al., 2020, Roy et al., 2021].

**Applicability.**    Before we proceed with the main development, we remark that the class of $\beta$-mixing is quite broad; a few examples where Theorem 3.1 can be instantiated are as follows:

- all $\phi$-mixing processes are $\beta$-mixing [Doukhan, 2012],
- stationary uniformly ergodic Markov chains are $\beta$-mixing,
- stationary Gaussian vector autoregressive moving average (ARMA) processes are $\beta$-mixing [Mokkadem, 1988],
- many other sub-classes of GARCH models, often studied in the economics and finance literature, are $\beta$-mixing [Carrasco and Chen, 2002].

The list is far from exhaustive and further examples can for instance be found in Doukhan [2012]. The stationarity assumptions above can also typically be dropped. We also point out that it is precisely because we can handle misspecification that our result is of interest for many of these examples.

**Outline.**    The rest of this article is structured as follows. Section 2 fixes our notation and yields a more formal problem formulation. We provide our main result, Theorem 3.1, in Section 3. After stating our main theorem, we highlight its features and then proceed to compare it to related work in Section 3.1. We outline the proof of Theorem 3.1 and provide supporting results in Section 4, including separate analyses of the noise-interaction and the lower tail of the empirical covariance matrix. Section 5 concludes and technical details are relegated to Appendix A.

## 2   Preliminaries

**Notation.**    Expectation (resp. probability) with respect to all the randomness of the underlying probability space is denoted by $\mathbf{E}$ (resp. $\mathbf{P}$). For two probability measures $\mathsf{P}$ and $\mathsf{Q}$ defined on the same probability space, their total variation is denoted $\|\mathsf{P} - \mathsf{Q}\|_{\mathsf{TV}}$. Maxima (resp. minima) of two numbers $a, b \in \mathbb{R}$ are denoted by $a \vee b = \max(a, b)$ (resp. $a \wedge b = \min(a, b)$). For an integer $n \in \mathbb{N}$, we also define the shorthand $[n] \triangleq \{1, \ldots, n\}$.

The Euclidean norm on $\mathbb{R}^d$ is denoted $\|\cdot\|_2$, and the unit sphere in $\mathbb{R}^d$ is denoted $\mathbb{S}^{d-1}$. The standard inner product on $\mathbb{R}^d$ is denoted $\langle \cdot, \cdot \rangle$. We embed matrices $M \in \mathbb{R}^{d_1 \times d_2}$ in Euclidean space by vectorization: $\operatorname{vec} M \in \mathbb{R}^{d_1 d_2}$, where vec is the operator that vertically stacks the columns of $M$ (from left to right and from top to bottom). For a matrix $M$, the Euclidean norm is the Frobenius norm, i.e., $\|M\|_F \triangleq \|\operatorname{vec} M\|_2$. We similarly define the inner product of two matrices $M, N$ by $\langle M, N \rangle \triangleq \langle \operatorname{vec} M, \operatorname{vec} N \rangle$. The transpose of a matrix $M$ is denoted by $M^{\mathsf{T}}$—and if $M$ is square—$\operatorname{tr} M$ denotes its trace. We also write $\|M\|_{\mathsf{op}}$ for the induced $(\mathbb{R}^d, \|\cdot\|_2) \to (\mathbb{R}^d, \|\cdot\|_2)$ norm. For two symmetric matrices $M, N$, we write $M \succ N$ ($M \succeq N$) if $M - N$ is positive (semi-)definite. If $M$ is positive semidefinite we write $\partial M \triangleq \{x \in \mathbb{R}^d \mid x^{\mathsf{T}} M x = 1\}$ for the boundary of the ellipsoid induced by $M$ (note that $\mathbb{S}^{d-1} = \partial I_d$).

**Problem Formulation.**    We are given $n$ input-output tuples: $X_{1:n} \sim \mathsf{P}^X_{1:n}$ (taking values in $\mathbb{R}^{d_{\mathsf{X}}}$) and $Y_{1:n} \sim \mathsf{P}^Y_{1:n}$ (taking values in $\mathbb{R}^{d_{\mathsf{Y}}}$). Using these samples, the goal of the learner is to estimate the best linear hypothesis:

$$M_\star \in \underset{M \in \mathbb{R}^{d_{\mathsf{Y}} \times d_{\mathsf{X}}}}{\operatorname{argmin}} \left\{ \mathbf{E} \|Y - MX\|_2^2 \right\} \tag{2.1}$$

where the distributions of $X$ and $Y$ in (2.1) are specified via:

$$X \sim \frac{1}{n} \sum_{i=1}^{n} \mathsf{P}^X_i \qquad \text{and} \qquad Y \sim \frac{1}{n} \sum_{i=1}^{n} \mathsf{P}^Y_i. \tag{2.2}$$

Note that (2.2) is equivalent to sampling from the uniform mixture over $(X_{1:n}, Y_{1:n})$ with the index $i \in [n]$ sampled uniformly. The operator $\Sigma_X \triangleq \mathbf{E}[XX^\mathsf{T}]$ is the averaged covariance operator (with $X$ as in (2.2)). The excess risk of a linear hypothesis $M$ can then be written as:

$$\text{Excess Risk}\,(M) \triangleq \mathbf{E}\|Y - MX\|_2^2 - \mathbf{E}\|Y - M_\star X\|_2^2 = \|(M - M_\star)\sqrt{\Sigma_X}\|_F^2. \qquad (2.3)$$

We now define the *noise variable* $W_i \triangleq Y_i - M_\star X_i$ but, as mentioned above, do not impose any (conditional) mean zero assumptions on the noise. To simplify the exposition, we will henceforth assume that $\Sigma_X \succ 0$, but our results easily extend to the case $\Sigma_X \succeq 0$ by restricting attention to the span of $\Sigma_X$. With these preliminaries in place, on the event that the design is nondegenerate, the OLS and its error equation can be specified as follows:

$$\widehat{M} \triangleq \left(\sum_{i=1}^n Y_i X_i^\mathsf{T}\right)\left(\sum_{i=1}^n X_i X_i^\mathsf{T}\right)^{-1} \implies \widehat{M} - M_\star = \left(\sum_{i=1}^n W_i X_i^\mathsf{T}\right)\left(\sum_{i=1}^n X_i X_i^\mathsf{T}\right)^{-1}. \qquad (2.4)$$

Our task in the sequel is to establish that the choice $\widehat{M}$ renders the excess risk (2.3) small. We note in passing that $\widehat{M}$ is an empirical risk minimizer:

$$\widehat{M} \in \operatorname*{argmin}_{M \in \mathbb{R}^{d_Y \times d_X}} \left\{\frac{1}{n}\sum_{i=1}^n \|Y_i - MX_i\|_2^2\right\}.$$

**The Noise Term.** Let us also define the following prefiltered noise-class interaction variables:

$$V_i \triangleq W_i X_i^\mathsf{T} \Sigma_X^{-1/2} \qquad i \in [n]. \qquad (2.5)$$

The square of the following (weighted and possibly biased) random walk effectively characterizes the noise level in our problem:

$$S_n \triangleq \frac{1}{n}\sum_{i=1}^n V_i. \qquad (2.6)$$

We remark that by construction $\mathbf{E}S_n = 0$ using the optimality of $M_\star$ in (2.1). To see this, simply invoke the optimality equation for $M_\star$ and note that $\mathbb{R}^{d_X \times d_Y}$ induces a convex class in the corresponding $L^2$-space over the mixtures (2.2). Note however that the increments of (2.6) are not necessarily mean zero unless $X_{1:n}$ and $Y_{1:n}$ are stationary. However, since $\mathbf{E}S_n = 0$, we also have with $\bar{V}_i \triangleq V_i - \mathbf{E}V_i$:

$$S_n = \frac{1}{n}\sum_{i=1}^n \bar{V}_i. \qquad (2.7)$$

In light of (2.4) and (2.6), we have that the empirical excess risk depends on the norm of:

$$(\widehat{M} - M_\star)\sqrt{\Sigma_X} = S_n \left(\frac{1}{n}\sum_{i=1}^n \Sigma_X^{-1/2} X_i X_i^\mathsf{T} \Sigma_X^{-1/2}\right)^{-1}. \qquad (2.8)$$

Hence, we need to control the random walk in (2.7) and the lower tail of

$$\tilde{\Sigma}_n \triangleq \frac{1}{n}\sum_{i=1}^n \Sigma_X^{-1/2} X_i X_i^\mathsf{T} \Sigma_X^{-1/2}, \qquad (2.9)$$

the prefiltered empirical covariance matrix. As mentioned previously, lower uniform laws for (2.9) are valid under mild assumptions [Koltchinskii and Mendelson, 2015, Oliveira, 2016], and blocking such results does not incur more than a worsening of the burn-in. Hence, the noise level of the problem is very much dictated by the random walk (2.6).

$\beta$**-mixing and the Blocking Technique.** In the sequel we demonstrate that the standard blocking device combined with a (functional) version of Bernstein's inequality allows us to pass the distributional (or coarse) measure of dependency to a higher order additive term, yielding non-asymptotic rates consistent with the CLT as described in Section 1. We will also use blocking to derive our lower uniform law, controlling the lower tail of (2.9). To make these ideas rigorous we require the following standard measure of dependence (take $Z_{1:n} = (X, Y)_{1:n}$ below).

**Definition 2.1.** *Let $Z_{1:n}$ be a stochastic process. The $\beta$-mixing coefficients of $Z$, $\beta_Z(i)$, are:*

$$\beta_Z(i) = \sup_{t \in [n]: t+i \leq n} \mathbf{E} \| \mathsf{P}_{Z_{i+t}}(\cdot \mid Z_{1:t}) - \mathsf{P}_{Z_{i+t}} \|_{\mathsf{TV}}, \quad i \in [n]. \tag{2.10}$$

Intuitively, the coefficients $\beta_Z(i)$ in (2.10) measure the dependence at range $i$ of the process $Z_{1:n}$. This measurement is done in total variation distance by comparing the distribution of $Z_{t+i}$ with the conditional distribution $Z_{t+i} \mid Z_{1:t}$. More concretely, the notion of $\beta$-mixing allows us to use the blocking technique Yu [1994]. This technique splits the process $Z_{1:n}$ into blocks, such that every other block is approximately independent, leaving us with two separate processes, consisting of odd and even blocks, that are almost independent (see (4.1) below). One then proceeds to use $\beta$-mixing to construct a "parallel" probability space, approximating the original one, but in which the odd and even blocks *are independent*. The price we pay for this is measured in terms of the coefficients (2.10). The essence of this idea—to use that data points sufficiently separated in time are often roughly independent—can be traced back to Bernstein [1927].

## 3 Main Result

To give our main result for general target dimension we require one last preliminary notion. Given a $d$-dimensional square, symmetric positive semidefinite matrix $M \in \mathbb{R}^{d \times d}$, we say that its *effective dimension* is $\mathsf{edim}(M) \triangleq \operatorname{tr} M / \|M\|_{\mathsf{op}}$. Our main result is the following theorem.

**Theorem 3.1.** *Fix $\delta \in (0,1)$ and $n, m \in \mathbb{N}$ with $2m \leq n$. Let $a_{1:2m}$ be a monotone partition of $[n]$ such that if $k \in a_i$, $l \in a_j$, and $i > j$, then $k > l$ holds. Set $a_{\max} \in \operatorname{argmax}_{i \in [2m]} |a_i|$. Fix also a $\beta$-mixing sequence $(X, Y)_{1:n}$ of which each element admits at least $s \in [4, \infty)$ moments. Assume that there exists a positive number $\mathsf{h} \in \mathbb{R}$ such that for every $v \in \partial \Sigma_X$ and $i \in [n]$ we have that $\mathbf{E} \langle v, X_i \rangle^4 \leq \mathsf{h}^2 \langle v, \mathbf{E}[X_i X_i^{\mathsf{T}}] v \rangle$. Define also the noise interaction terms:*

$$\Sigma \triangleq \frac{1}{n} \sum_{i=1}^{2m} \Sigma_i, \quad \Sigma_i \triangleq \mathbf{E} \left[ \mathsf{vec} \left( \sum_{j \in a_i} \bar{V}_j \right) \mathsf{vec} \left( \sum_{j \in a_i} \bar{V}_j \right)^{\mathsf{T}} \right], \quad \sigma^2 \triangleq \|\Sigma\|_{\mathsf{op}}. \tag{3.1}$$

*There exist universal positive constants $c_1, c_2, c_3, c_4, c_5, c_6$ such that with probability at least $1 - \delta$:*

$$\|(\widehat{M} - M_\star)\sqrt{\Sigma_X}\|_F^2 \leq \frac{c_1 \sigma^2 (\mathsf{edim}(\Sigma) + \log(1/\delta))}{n} \tag{3.2}$$

*as long as the following burn-in conditions hold:*

$$\frac{n}{|a_{\max}|} \geq c_2(d_{\mathsf{X}} + \mathsf{h}^2 \log(1/\delta)); \quad \left( \frac{n}{|a_{\max}|} \right)^{1-2/s} \geq c_3 s^2 \frac{\left( \frac{1}{m} \sum_{i=1}^{2m} \mathbf{E} \left\| \frac{1}{\sqrt{|a_{\max}|}} \sum_{j \in a_i} \bar{V}_j \right\|_F^s \right)^{2/s}}{\mathsf{edim}(\Sigma) \sigma^2 \delta^{2/s}}; \tag{3.3}$$

$$c_4^{-1} < \frac{\sum_{i \in [2m] \cap 2\mathbb{N}} |a_i|}{\sum_{i \in [2m] \cap (2\mathbb{N}-1)} |a_i|} < c_4; \quad \sum_{i \in [2m] \cap (2\mathbb{N}-1)} c_5^{-1} \Sigma_i \preceq \sum_{i \in [2m] \cap 2\mathbb{N}} \Sigma_i \preceq \sum_{i \in [2m] \cap (2\mathbb{N}-1)} c_5 \Sigma_i; \tag{3.4}$$

$$\sum_{i=2}^{2m-1} \beta_{X,Y}(|a_i|) \leq c_6 \delta; \tag{3.5}$$

*and where $\beta_{X,Y}$ are the $\beta$-mixing coefficients of $(X, Y)_{1:n}$.*

To interpret Theorem 3.1 we proceed with a sequence of remarks, discussing its features. These remarks also serve to parse the terminology above and lead us to a simplified statement for stationary data and 1-dimensional targets. We present this simplified version as Corollary 3.1 below.

- The dimensional scaling in (3.2) is captured by the effective dimension term $\mathsf{edim}(\Sigma)$, which always lies in the interval $[1, d_{\mathsf{X}} d_{\mathsf{Y}}]$. For one-dimensional targets and benign noise interaction, and since the $X_i$ in the noise term $V_i$ have been whitened (see (2.5)), we expect $\Sigma$ to be roughly isotropic. Indeed, whenever $d_{\mathsf{Y}} = 1$, the trivial bound $\mathsf{edim}(\Sigma) \leq d_{\mathsf{X}}$ produces the familiar behavior: $\|(\widehat{M} - M_\star)\sqrt{\Sigma_X}\|_2^2 \lesssim \sigma^2 d_{\mathsf{X}}/n$ with high probability.

- The scaling with $d_X, d_Y, \sigma^2$ and $n$ thus scales as expected in the iid regime, but also degrades gracefully with dependence. We reiterate that (3.2) does not depend directly on mixing in the leading order term. For comparison, under suitable regularity conditions, the noise term predicted by the CLT is:

$$\Sigma_{\text{CLT}} \triangleq \limsup_{n \to \infty} n^{-1} \mathbf{E} \left[ \text{vec} \left( \sum_{j=1}^{n} \bar{V}_j \right) \text{vec} \left( \sum_{j=1}^{n} \bar{V}_j \right)^{\mathsf{T}} \right], \quad (3.6)$$

and one can achieve $\Sigma = \Sigma_{\text{CLT}} + o(1)$ in most situations of practical interest by tuning the block-length; therefore, our analysis of OLS essentially matches the optimal asymptotics.

- Moreover, the constant $c_1$ appearing in (3.2) is quite benign and can be made arbitrarily close to 2 by suitably inflating our burn-in constants $c_2, c_3, c_4, c_5, c_6$. We have however not been able to approach the optimal leading constant 1 in front of $\text{edim}(\Sigma)$. This can be traced to an application of the triangular inequality in Corollary 4.1.

- The moment bound $\mathbf{E}\langle v, X_i \rangle^4 \leq h^2 \langle v, \mathbf{E}[X_i X_i^{\mathsf{T}}] v \rangle$ $(v \in \partial \Sigma_X)$ is easily satisfied for e.g., Gaussian or bounded processes but is of course much milder than either assumption. The assumption that $s \geq 4$ can be relaxed to $s > 2$ by replacing our result controlling the lower tail, Theorem 4.3. We have chosen to present our result for $s \geq 4$ to strike a balance between expositional clarity and generality.

- The first burn-in condition of (3.3) is standard for control of the lower tail—beside the deflation factor $|a_{\max}|$, it is necessary even for iid data to guarantee that the "denominator" (2.9) is nonsingular. The second condition of (3.3)—in which a ratio of $s$:th and 2nd moment of the noise variable appears—is the price we pay in the moderate deviations bandwidth for only having $s$ moments: it controls the rate at which the random walk (2.6) approaches asymptotic normality and reduces to the condition of Oliveira [2016] in the iid regime. This latter condition can in principle be removed with slightly modified constants if sufficiently many moments of the data-generating process satisfy a sub-Gaussian type moment equivalence condition (in which case the above-mentioned ratio is constant). Without such an assumption, we note that some polynomial dependence on $1/\delta$ is necessary under our tail assumptions and is not an artifact of our analysis; OLS is not deviation-optimal in the entire range of $\delta \in (0,1)$—due to the presence of the random walk (2.6) in the numerator—unless the noise variables have Gaussian-like tails [cf. Mendelson, 2018, Section 6.4].

- The conditions in (3.4) and (3.5) relate to dependence. The last condition (3.5) simply asks that our process mixes sufficiently fast. If the mixing coefficients $\beta_{X,Y}(|a_i|)$ are exponential, this amounts to a logarithmic burn-in in $1/\delta$. However, we can still handle slow, polynomial mixing rates, at the cost of a polynomial burn-in in $1/\delta$. The conditions in (3.4) asks that the odd and even blocks are balanced in terms of their length and second order statistics. It is trivially satisfied for (weakly) stationary processes analyzed using a uniform blocking length (length of the $a_i$).

In light of the above remarks, we are now in position to simplify Theorem 3.1. If we impose stationarity, quite a few terms in the burn-in conditions (3.3),(3.4) and (3.5) either simplify or vanish. Further restricting to the case where targets are 1-dimensional and letting the sample-size be divisible by the block-length yields the corollary below.

**Corollary 3.1.** *Fix $\delta \in (0,1)$ and $n, \tau \in \mathbb{N}$ and let $2\tau$ divide $n$. Fix also a joint distribution of 1-dimensional targets and $d_X$-dimensional covariates $\mathsf{P}^{X,Y}$ with at least $s \in [4, \infty)$ moments. Let $(X,Y)_{1:n}$ be a stationary $\beta$-mixing sequence with marginals equal to $\mathsf{P}^{X,Y}$. Assume further that there exists a positive number $h \in \mathbb{R}$ such that for every $v \in \partial \Sigma_X$ we have that $\mathbf{E}\langle v, X \rangle^4 \leq h^2$ where $X \sim \mathsf{P}^X$. Define also the noise interaction term: $\sigma^2 \triangleq \frac{1}{\tau} \sup_{v \in \mathbb{S}^{d_X-1}} \mathbf{E}\left[ \left( \sum_{i=1}^{\tau} \langle \bar{V}_i, v \rangle^2 \right) \right]$. There exist universal positive constants $c_1, c_2, c_3, c_4$ such that with probability at least $1 - \delta$:*

$$\|(\widehat{M} - M_\star) \sqrt{\Sigma_X}\|_2^2 \leq \frac{c_1 \sigma^2 (d_X + \log(1/\delta))}{n}$$

*as long as the following burn-in conditions hold:*

$$\frac{n}{\tau} \geq c_2 (d_X + h^2 \log(1/\delta)); \quad \left( \frac{n}{\tau} \right)^{1-2/s} \geq c_3 s^2 \frac{\left( \mathbf{E} \left\| \frac{1}{\sqrt{\tau d_X}} \sum_{i=1}^{\tau} \bar{V}_i \right\|_2^s \right)^{2/s}}{\sigma^2 \delta^{2/s}}; \quad \frac{n}{\tau} \beta_{X,Y}(\tau) \leq c_4 \delta. \quad (3.7)$$

Corollary 3.1 takes a very similar form to—by now—standard results in the iid regime [Hsu et al., 2012, Oliveira, 2016]. Indeed, if the data is stationary the price we pay for dependence is that:

- variance and moment terms need to be computed in blocks;
- the burn-in is deflated by a factor of the block-length (the first two parts of (3.7)); and
- we incur an additional burn-in penalizing slow mixing—the last part of (3.7) asks that the block-length is not "too small".

## 3.1 Comparison to Related Work

Having established our main result, Theorem 3.1, we now provide a more detailed comparison to the relevant literature. Most closely related to our results is Nagaraj et al. [2020], who study bounded linear regression models in which the data comes from an exponentially ergodic Markov chain. They find that strictly realizable linear regression is no harder than its iid counterpart in this setting, and show that a parallelized gradient algorithm achieves the optimal rate. More interestingly, in the absence of realizability, they also establish a lower bound demonstrating that the worst-case (global minimax) excess risk across all Markov chains with a given mixing time is deflated by said mixing time, thereby establishing a gap between realizable and non-realizable learning from dependent data.

Of course, their lower bound is no longer valid if one drops the requirement that the predictor performs uniformly well across all distributions with a prescribed mixing time. It is exactly herein that our analyses differ. While Nagaraj et al. [2020] characterize the worst-case (or global) complexity of linear regression, we focus on the instance-specific (or local) complexity. In other words, they compete against the worst distribution at a given level of mixing, whereas we compete against a fixed distribution. To appreciate this distinction, let us momentarily assume that $d_{\mathsf{X}} = d_{\mathsf{Y}} = 1$. The noise term $\sigma^2$ in Theorem 3.1 can be upper-bounded as:

$$\sigma^2 = \frac{1}{n} \sum_{i=1}^{2m} \mathbf{E}\left[\left(\sum_{j \in a_i} \bar{V}_j\right)^2\right] \leq \max_{i \in [2m]} |a_i| \cdot \max_{i \in [n]} \mathbf{E}\bar{V}_i^2 \tag{3.8}$$

by the Cauchy-Schwarz inequality. The right hand side of (3.8) is precisely inflated by the (maximal) block-length $\max_{i \in [2m]} |a_i|$.

Seen in this light, our results being sharper in terms of the measure of dependency reduces to stating that our results are sharper by an application of the Cauchy-Schwarz Inequality. Moreover, the statement that the global complexity is worse than its iid counterpart by a factor of the mixing time amounts in our setting to stating that there exists a distribution achieving equality in (3.8). We remark that such a distribution is easily constructed by taking $X_{1:n}$ and $Y_{1:n}$ to be constant within each block and stationary across the blocks; this is precisely when the application of the Cauchy-Schwarz inequality in (3.8) turns to equality. To further appreciate the distinction between our results, note that our result measures dependence through correlation. By contrast, a result scaling with the mixing time measures dependence in a stronger variational sense. That is, the former measures dependence at the level of orthogonality of the random variables themselves, whereas the latter measures it at the level of orthogonality of all measurable functions of these random variables.

**Further Related Work.** Another closely related line of work studies parameter identification in auto-regressive models [for a recent survey, see Tsiamis et al., 2022]. When the noise model is strictly realizable—the variables $W_{1:n}$ form a martingale difference sequence with respect to the filtration generated by $X_{1:n}$—identification is possible at the iid rate even in the absence of mixing [Simchowitz et al., 2018, Faradonbeh et al., 2018, Sarkar and Rakhlin, 2019]. Naturally, our results do not cover the mixing-free regime as we consider: (1) the agnostic setting in which self-normalized martingale arguments [Peña et al., 2009, Abbasi-Yadkori et al., 2011] are not available; and (2) excess risk bounds instead of parameter identification—it seems unlikely that (3.2) holds without some notion of stochastic stability due to the presence of $\Sigma_X$ on the left hand side.

More generally—moving beyond linear time-series models—several authors have considered learning under various weak dependency notions. Kuznetsov and Mohri [2017] give generalization bounds in a more general setting using the same blocking technique—due to Yu [1994]—used here. Statements similar in spirit can also be found in e.g., Steinwart and Christmann [2009], Duchi et al. [2012] and

most recently Roy et al. [2021]. However, they all suffer the dependency deflation discussed above and in our introduction (Section 1). We also note that Ziemann and Tu [2022] obtain rates for strictly realizable square loss that—similar to ours here—relegate mixing times into additive burn-in factors. While they treat more general hypothesis classes, they do not go beyond strict realizability, and their analysis rests on the assumption that the noise interaction term is a martingale difference sequence.

## 4 Proof Overview

Theorem 3.1 is the direct consequence of two separate results: Theorem 4.2, which controls the centered noise (2.7) term, and Theorem 4.3, which bounds the lower tail of the normalized empirical covariance matrix (2.9). We prove Theorem 4.2 by blocking the Fuk-Nagaev inequality of Einmahl and Li [2008], and Theorem 4.3 by a truncation argument combined with blocking. These results are found in Section 4.2 and Section 4.3. Our proof idea is heavily inspired by that of Oliveira [2016] and the idea is very much to adjust his approach in such a way that blocking does not affect the leading term in the rate.[2] As either result relies on blocking, it is now pertinent to describe this technique in a little more detail.

### 4.1 Blocking

Recall that we partition $[n]$ into $2m$ consecutive intervals, denoted $a_j$ for $j \in [2m]$, so that $\sum_{j=1}^{2m} |a_j| = n$. Denote further by $O$ (resp. by $E$) the union of the oddly (resp. evenly) indexed subsets of $[n]$. We further abuse notation by writing $\beta_Z(a_i) = \beta_Z(|a_i|)$ in the sequel.

We split the process $Z_{1:n}$ as:

$$Z^o_{1:|O|} \triangleq (Z_{a_1}, \ldots, Z_{a_{2m-1}}), \quad Z^e_{1:|E|} \triangleq (Z_{a_2}, \ldots, Z_{a_{2m}}). \tag{4.1}$$

Let $\tilde{Z}^o_{1:|O|}$ and $\tilde{Z}^e_{1:|E|}$ be blockwise decoupled versions of (4.1). That is we posit that $\tilde{Z}^o_{1:|O|} \sim \mathsf{P}_{\tilde{Z}^o_{1:|O|}}$ and $\tilde{Z}^e_{1:|E|} \sim \mathsf{P}_{\tilde{Z}^e_{1:|E|}}$, where:

$$\mathsf{P}_{\tilde{Z}^o_{1:|O|}} \triangleq \mathsf{P}_{Z_{a_1}} \otimes \mathsf{P}_{Z_{a_3}} \otimes \cdots \otimes \mathsf{P}_{Z_{a_{2m-1}}} \quad \text{and} \quad \mathsf{P}_{\tilde{Z}^e_{1:|E|}} \triangleq \mathsf{P}_{Z_{a_2}} \otimes \mathsf{P}_{Z_{a_4}} \otimes \cdots \otimes \mathsf{P}_{Z_{a_{2m}}}. \tag{4.2}$$

The process $\tilde{Z}_{1:n}$ with the same marginals as $\tilde{Z}^o_{1:|O|}$ and $\tilde{Z}^e_{1:|E|}$ is said to be the decoupled version of $Z_{1:n}$. To be clear: $\mathsf{P}_{\tilde{Z}_{1:n}} \triangleq \mathsf{P}_{Z_{a_1}} \otimes \mathsf{P}_{Z_{a_2}} \otimes \cdots \otimes \mathsf{P}_{Z_{a_{2m}}}$, so that $\tilde{Z}^o_{1:|O|}$ and $\tilde{Z}^e_{1:|E|}$ are alternatingly embedded in $\tilde{Z}_{1:n}$. The following result is key—by skipping every other block, $\tilde{Z}_{1:n}$ may be used in place of $Z_{1:n}$ for evaluating scalar functions at the cost of an additive mixing-related term.

**Proposition 4.1** (Lemma 2.6 in Yu [1994]; Proposition 1 in Kuznetsov and Mohri [2017]). *Fix a $\beta$-mixing process $Z_{1:n}$ and let $\tilde{Z}_{1:n}$ be its decoupled version. For any measurable function $f$ of $Z^o_{1:|O|}$ (resp. $g$ of $Z^e_{1:|E|}$) with joint range $[0, 1]$ we have that:*

$$
\begin{aligned}
|\mathbf{E}(f(Z^o_{1:|O|})) - \mathbf{E}(f(\tilde{Z}^o_{1:|O|}))| &\leq \sum_{i \in E \setminus \{2m\}} \beta_Z(a_i), \\
|\mathbf{E}(g(Z^e_{1:|E|})) - \mathbf{E}(g(\tilde{Z}^e_{1:|E|}))| &\leq \sum_{i \in O \setminus \{1\}} \beta_Z(a_i).
\end{aligned}
\tag{4.3}
$$

The following corollary to Proposition 4.1 is convenient for controlling norms of random walks.

**Corollary 4.1** (Lemma 3 in Kuznetsov and Mohri [2017]). *Let $Z_{1:n}$ be a $\beta$-mixing process taking values in a normed space $(\mathsf{Z}, \|\cdot\|)$, and let $\tilde{Z}_{1:n}$ be its decoupled version. For any*

---

[2]At a high level, for iid data, this proof strategy first appeared in the journal version of Oliveira [2016].

$\varepsilon \geq \mathbf{E} \left\| \frac{1}{|O|} \sum_{i \in O} \tilde{Z}_i \right\| \vee \mathbf{E} \left\| \frac{1}{|E|} \sum_{i \in E} \tilde{Z}_i \right\|$ *we have that:*

$$\mathbf{P} \left( \left\| \frac{1}{n} \sum_{i=1}^{n} Z_i \right\| > \varepsilon \right) \leq \sum_{i=1}^{2m} \beta_Z(a_i)$$

$$+ \mathbf{P} \left( \left\| \frac{1}{|O|} \sum_{i \in O} \tilde{Z}_i \right\| > \mathbf{E} \left\| \frac{1}{|O|} \sum_{i \in O} \tilde{Z}_i \right\| + \varepsilon_o \right) + \mathbf{P} \left( \left\| \frac{1}{|E|} \sum_{i \in E} \tilde{Z}_i \right\| > \mathbf{E} \left\| \frac{1}{|E|} \sum_{i \in E} \tilde{Z}_i \right\| + \varepsilon_e \right),$$
(4.4)

*where* $\varepsilon_o = \varepsilon - \mathbf{E} \left\| \frac{1}{|O|} \sum_{i \in O} \tilde{Z}_i \right\|$ *and* $\varepsilon_e = \varepsilon - \mathbf{E} \left\| \frac{1}{|E|} \sum_{i \in E} \tilde{Z}_i \right\|$.

In short, up to a mild failure additional failure probability term, we only need to control the tensor product processes (4.2).

## 4.2 Dependent Random Walks

Once equipped with Corollary 4.1, we still require control of the independent blocks. The following Fuk-Nagaev inequality due to Einmahl and Li [2008] provides such control.

**Theorem 4.1** (Theorem 4 in Einmahl and Li [2008]). *Fix* $s > 2$, *a separable normed space* $(\mathsf{U}, \| \cdot \|)$ *and a* $\mathsf{U}$*-valued sequence* $U_{1:n}$ *of independent random variables. Assume that* $\mathbf{E}\|U_i\|^s < \infty$ *for* $i \in [n]$. *Then for any* $\varepsilon \in (0, \infty)$, $\eta \in (0, 1]$, *and* $t \geq 0$, *we have that:*

$$\mathbf{P} \left( \max_{k \in [n]} \left\| \sum_{i=1}^{k} U_i \right\| \leq (1+\eta)\mathbf{E} \left\| \sum_{i=1}^{n} U_i \right\| + (1+9\varepsilon)t \right)$$

$$\leq \exp \left( -\frac{t^2}{(2+\eta)\Lambda} \right) + C_{\varepsilon,\eta,s} \sum_{i=1}^{n} \frac{\mathbf{E}\|U_i\|^s}{t^s}, \quad (4.5)$$

*where* $\Lambda \triangleq \sup_{v \in \mathscr{S}^*} \mathbf{E} \sum_{i=1}^{n} v^2(U_i)$ *and where* $\mathscr{S}^*$ *is unit disk in the dual space of* $(\mathsf{U}, \| \cdot \|)$. *Moreover, we may take* $C_{\varepsilon,\eta,s} = \left( 1 + (2s/e)^{2s}(2(1+2/\varepsilon)(3+4/\eta))^2 + \varepsilon^{-s} \right)$.[3]

If our data were drawn independently, Theorem 4.1 would give us the required control of the random walk (2.7). The right hand side of (4.5) consists of a *mixed tail*: (1) a sub-Gaussian term with a CLT-like weak variance term $\Lambda$; and (2) a polynomial term accounting for the fact that we only imposed the existence of $s$ moments. With the preliminary results Corollary 4.1 and Theorem 4.1 in place, we are now in position to control dependent random walks of the form (2.7).

**Theorem 4.2.** *Fix* $s > 2$, *a separable normed space* $(\mathsf{Z}, \| \cdot \|)$, *constants* $\varepsilon, \eta > 0$, *and set* $C_{\varepsilon,\eta,s} = \left( 1 + (2s/e)^{2s}(2(1+2/\varepsilon)(3+4/\eta))^2 + \varepsilon^{-s} \right)$. *Fix also a consecutive partition* $a_{1:2m}$ *of* $[n]$ *and let* $Z_{1:n}$ *be a mean zero,* $\beta$*-mixing process taking values in* $\mathsf{Z}$ *with block decoupled version* $\tilde{Z}_{1:n}$. *Let* $O$ *be the union of the odd* $a_i$ *and* $E$ *be the union of the even* $a_i$. *Assume that* $\mathbf{E}\|Z_i\|^s < \infty$ *for* $i \in [n]$. *For every* $\varepsilon, \eta > 0$ *and* $\delta \in (0, 1)$, *we have that:*

$$\mathbf{P} \left( \left\| \frac{1}{n} \sum_{i=1}^{n} Z_i \right\| \geq \max_{\mathsf{sgn} \in \{O,E\}} \sqrt{\frac{\Lambda_{\mathsf{sgn}}}{|\mathsf{sgn}|}} \left( (1+2\eta)\sqrt{r} + (1+9\varepsilon)\sqrt{(2+\eta)\log(1/\delta)} \right) \right)$$

$$\leq 2\delta + \sum_{i=2}^{2m-1} \beta_Z(a_i) + \frac{C_{\varepsilon,\eta,s}(1+9\varepsilon)^s}{r^{s/2}\eta^s} \sum_{\mathsf{sgn} \in \{O,E\}} \sum_{i \in [2m]: a_i \subset \mathsf{sgn}} \frac{\mathbf{E} \left\| \sum_{j \in a_i} Z_j \right\|^s}{|\mathsf{sgn}|^{s/2} \Lambda_{\mathsf{sgn}}^{s/2}}, \quad (4.6)$$

*where* $\Lambda_{\mathsf{sgn}} \triangleq \sup_{v \in \mathscr{S}^*} \frac{1}{|\mathsf{sgn}|} \sum_{a_i \subset \mathsf{sgn}} \mathbf{E} v^2 \left( \sum_{j \in a_i} Z_j \right)$ *for* $\mathsf{sgn} \in \{O, E\}$, $\mathscr{S}^*$ *is unit disk in the*

*dual space of* $(\mathsf{Z}, \| \cdot \|)$, *and* $\sqrt{r} \geq \max_{\mathsf{sgn} \in \{O,E\}} \dfrac{\mathbf{E} \left\| \frac{1}{\sqrt{|\mathsf{sgn}|}} \sum_{i \in \mathsf{sgn}} \tilde{Z}_i \right\|}{\sqrt{\Lambda_{\mathsf{sgn}}}}$.

---

[3]The constant $C_{\varepsilon,\eta,s}$ is not specified exactly in Einmahl and Li [2008]. This constant is easy to obtain by observing that their $K_s$, which may be taken to be the best constant such that $(\log x)^{2s} \leq K_s x$ for $x \geq 1$, is upper-bounded by $(2s/e)^{2s}$.

In Theorem 4.2 we have combined the blocking technique with the Fuk-Nagaev inequality (4.5). The right hand side of (4.6) is exactly as in (4.5) but instantiated to our setting and with extra additive mixing-related term.

### 4.3 The Lower Tail of the Empirical Covariance Matrix

We now proceed to analyze the lower tail of the empirical covariance matrix (2.9).

**Theorem 4.3.** *Fix $\delta > 0$ and a consecutive partition $a_{1:2m}$ of $[n]$. Let $X_{1:n}$ be a sequence of $\beta$-mixing random variables taking values in $\mathbb{R}^{d_X}$ with finite fourth moment. Assume that there exists a positive number $\mathsf{h} \in \mathbb{R}$ such that for every $v \in \partial\Sigma_X$ and $i \in [n]$, we have $\mathbf{E}\langle v, X_i\rangle^4 \leq \mathsf{h}^2\langle v, \mathbf{E}[X_i X_i^\mathsf{T}]v\rangle$. There exists a positive universal constant $C \in \mathbb{R}$ such that as long as*

$$n \geq C \max_{j \in [2m]} |a_j|(d_X + \mathsf{h}^2 \log(1/\delta)) \quad \text{and} \quad \sum_{i=2}^{2m-1} \beta_X(a_i) \leq \frac{\delta}{2}, \tag{4.7}$$

*then*

$$\mathbf{P}\left(\forall v \in \mathbb{R}^{d_X} \ : \ \frac{1}{n}\sum_{i=1}^{n}\langle v, X_i\rangle^2 \geq \frac{1}{2n}\sum_{i=1}^{n}\mathbf{E}\langle v, X_i\rangle^2\right) \geq 1 - \delta. \tag{4.8}$$

It is by now a well-established fact that lower uniform laws of the form (4.8) hold under mild assumptions for various function classes (linear functions on $\mathbb{R}^{d_X}$ in this case). Since these assumptions are quite mild and only affect burn-in conditions, deflating the sample-size via blocking does not deflate the final convergence rate. The particular approach we have chosen here to establish Theorem 4.3 is to combine blocking with the approach found in [Wainwright, 2019, Theorem 14.12]. We remark that similar statements hold if one instead blocks the arguments of say Oliveira [2016] or Koltchinskii and Mendelson [2015].

## 5 Summary

The leading order term of our main result, Theorem 3.1, does not directly depend on any mixing-time type quantities. It mimics the asymptotic rate and scales solely in terms of the second order statistics of the process at hand. To arrive at this result, we rely on two facts:

- The lower tail of the empirical covariance matrix (2.9) is well-behaved under mild assumptions. In an excess risk bound, the contribution of the lower uniform law to the overall error is not of leading order. Hence, incurring a sample size deflation for this purpose is not critical.

- By combining blocking with a version of Bernstein's inequality, we are able to push the effect of blocking to only affect the large deviations regime. In the moderate and small deviations regimes, control of the leading order of the random walk in (2.7) is not directly impacted by slow mixing.

## Acknowledgements

Ingvar Ziemann is supported by a Swedish Research Council international postdoc grant. Nikolai Matni is supported in part by NSF award CPS-2038873, NSF CAREER award ECCS-2045834, and a Google Research Scholar award. The authors thank Bruce Lee for comments on an earlier draft of the manuscript.

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

# A Proofs

## A.1 Proof of Theorem 4.2

Let $\tilde{Z}_{i:n}$ be the block-decoupled version of $Z_{1:n}$. Let sgn $\in \{O, E\}$. For $i \in O$ we define $U_i = \frac{1}{|O|} \sum_{j \in a_i} \tilde{Z}_j$ and similarly for $i \in E$ we define $U_i = \frac{1}{|E|} \sum_{j \in a_i} \tilde{Z}_j$.

In light of Corollary 4.1 to control $\mathbf{P}\left(\left\|\frac{1}{n} \sum_{i=1}^n Z_i\right\| > u\right)$ it suffices to control

$$\mathbf{P}\left(\left\|\frac{1}{|\text{sgn}|} \sum_{i \in \text{sgn}} \tilde{Z}_i\right\| \leq \mathbf{E}\left\|\frac{1}{|\text{sgn}|} \sum_{i \in \text{sgn}} \tilde{Z}_i\right\| + u_{\text{sgn}}\right)$$

for both sgn $\in \{O, E\}$ and where $u_{\text{sgn}} = u - \mathbf{E}\left\|\frac{1}{|\text{sgn}|} \sum_{i \in \text{sgn}} \tilde{Z}_i\right\|$. Introduce now the change of variables $(1 + 9\varepsilon)t_{\text{sgn}} = u_{\text{sgn}} - \eta\left(\mathbf{E}\left\|\frac{1}{|\text{sgn}|} \sum_{i \in \text{sgn}} \tilde{Z}_i\right\|\right)$.

We invoke Theorem 4.1 twice; we do so once for the even blocks and once for the odd blocks. This yields for sgn $\in \{O, E\}$ that for any $\varepsilon \in (0, \infty), \eta \in (0, 1]$ and $t_{\text{sgn}} \geq 0$ we have that:

$$\mathbf{P}\left(\left\|\frac{1}{|\text{sgn}|} \sum_{i \in \text{sgn}} \tilde{Z}_i\right\| \leq (1 + \eta)\mathbf{E}\left\|\frac{1}{|\text{sgn}|} \sum_{i \in \text{sgn}} \tilde{Z}_i\right\| + (1 + 9\varepsilon)t_{\text{sgn}}\right)$$

$$\leq \exp\left(-\frac{|\text{sgn}|t_{\text{sgn}}^2}{(2 + \eta)\Lambda_{\text{sgn}}}\right) + C_{\varepsilon,\eta,s} \sum_{a_i \subset \text{sgn}} \frac{\mathbf{E}\left\|\sum_{j \in a_i} Z_j\right\|^s}{|\text{sgn}|^s t_{\text{sgn}}^s} \quad (\text{A.1})$$

with $\Lambda_{\text{sgn}}$ as announced and where we used that $\sum_{j \in a_i} Z_j$ is equal to $\sum_{j \in a_i} \tilde{Z}_j$ in distribution for every $a_i \subset [n]$.

Consequently, with $u = (1 + 9\varepsilon)t_{\text{sgn}} + (1 + \eta)\mathbf{E}\left\|\frac{1}{|\text{sgn}|} \sum_{i \in \text{sgn}} \tilde{Z}_i\right\|$ we find that (A.1) and Corollary 4.1 together yield that:

$$\mathbf{P}\left(\left\|\frac{1}{n} \sum_{i=1}^n Z_i\right\| \geq u\right) \leq \sum_{i=2}^{2m-1} \beta_Z(a_i)$$

$$+ \sum_{\text{sgn} \in \{O,E\}} \exp\left(-\frac{|\text{sgn}|t_{\text{sgn}}^2}{(2 + \eta)\Lambda_{\text{sgn}}}\right) + C_{\varepsilon,\eta,s} \sum_{a_i \subset \text{sgn}} \frac{\mathbf{E}\left\|\sum_{j \in a_i} Z_j\right\|^s}{|\text{sgn}|^s t_{\text{sgn}}^s} \quad (\text{A.2})$$

where summation over $a_i \subset$ sgn is restricted to $a_{1:2m}$. If for some $c > 0$ we choose

$$t_{\text{sgn}} \geq \max_{\text{sgn} \in \{O,E\}} \sqrt{\Lambda_{\text{sgn}}} \times \sqrt{\frac{c \vee (2 + \eta)\log(1/\delta)}{|\text{sgn}|}}$$

and let $\sqrt{r} \geq \max_{\text{sgn} \in \{O,E\}} \dfrac{\mathbf{E}\left\|\frac{1}{\sqrt{|\text{sgn}|}} \sum_{i \in \text{sgn}} \tilde{Z}_i\right\|}{\sqrt{\Lambda_{\text{sgn}}}}$ then (A.2) reads for either sgn $\in \{O, E\}$:

$$\mathbf{P}\left(\left\|\frac{1}{n} \sum_{i=1}^n Z_i\right\| \geq \max_{\text{sgn} \in \{O,E\}} \sqrt{\frac{\Lambda_{\text{sgn}}}{|\text{sgn}|}}\left((1 + \eta)\sqrt{r} + (1 + 9\varepsilon) \times \sqrt{c \vee (2 + \eta)\log(1/\delta)}\right)\right)$$

$$\leq \sum_{i=1}^{2m} \beta_Z(a_i) + 2\delta + \frac{C_{\varepsilon,\eta,s}}{c^{s/2}|O|^{s/2}} \sum_{a_i \subset O} \frac{\mathbf{E}\left\|\sum_{j \in a_i} Z_j\right\|^s}{\Lambda_O^{s/2}} + \frac{C_{\varepsilon,\eta,s}}{c^{s/2}|E|^{s/2}} \sum_{a_i \subset E} \frac{\mathbf{E}\left\|\sum_{j \in a_i} Z_j\right\|^s}{\Lambda_E^{s/2}}.$$

The result follows if we choose $c = r\eta^2/(1 + 9\varepsilon)^2$. $\blacksquare$

## A.2 Proof of Theorem 4.3

We will argue by truncation. This rests on the observation that for any sequence $F_{1:n}$ of measurable sets:

$$\sum_{i \in a} X_i X_i^\mathsf{T} \succeq \sum_{i \in a} X_i X_i^\mathsf{T} \mathbf{1}_{F_i}. \tag{A.3}$$

where as before we let $a \subset [n]$. The idea is that once we truncate we do not have to worry about the upper tail of the empirical covariance matrix. We begin by showing below that a well-chosen level of truncation still preserves most of the mass of the lower tail.

### A.2.1 Truncation step

We will want to choose the events $F_i$ to be constant across each block. We require the following simple observation. It states that the moment condition $\mathbf{E}\langle v, X_i \rangle^4 \leq \mathsf{h}^2 \langle v, \mathbf{E}[X_i X_i^\mathsf{T}]v \rangle$ extends naturally to the situation where each block is our basic unit of randomness.

**Lemma A.1.** *Under the hypotheses of Theorem 4.3 for every subset $a \subset [n]$ we have that:*

$$\mathbf{E}\left( \frac{1}{|a|} \sum_{i \in a} \langle v, X_i \rangle^2 \right)^2 \leq \mathsf{h}^2 \left( \frac{1}{|a|} \sum_{i \in a} \mathbf{E}\langle v, X_i \rangle^2 \right).$$

*Proof.* By direct calcuation:

$$\begin{aligned}
\mathbf{E}\left( \sum_{i \in a} \langle v, X_i \rangle^2 \right)^2 &\leq \left( \sum_{i \in a} \sqrt{\mathbf{E}\langle v, X_i \rangle^4} \right)^2 && \text{(Cauchy-Schwarz)} \\
&\leq \mathsf{h}^2 \left( \sum_{i \in a} \sqrt{\mathbf{E}\langle v, X_i \rangle^2} \right)^2 && \left( \mathbf{E}\langle v, X_i \rangle^4 \leq \mathsf{h}^2 \langle v, \mathbf{E}[X_i X_i^\mathsf{T}]v \rangle \right) \quad \text{(A.4)} \\
&\leq \mathsf{h}^2 |a| \sum_{i \in a} \mathbf{E}\langle v, X_i \rangle^2. && \text{(Cauchy-Schwarz)}
\end{aligned}$$

To finish the proof, divide both sides of (A.4) by $|a|^2$. ∎

As announced above, let now the $F_i$ be constant on each block $a \subset [n]$. In what follows, we abuse notation in the obvious way and write $F_a = F_i$ for any $i \in a$. The next lemma shows that suitable truncation preserves most of the mass of the lower tail.

**Lemma A.2.** *Impose the hypotheses of Theorem 4.3, fix $\tau > 0$ and let $F_a \triangleq \left\{ \frac{1}{|a|} \sum_{i \in a} \langle v, X_i \rangle^2 \leq \tau^2 \right\}$. For every subset $a \subset [n]$ we have that:*

$$\left( 1 - \frac{\mathsf{h}^2}{\tau^2} \right) \mathbf{E}\left[ \sum_{i \in a} \langle v, X_i \rangle^2 \right] \leq \sum_{i \in a} \mathbf{E}\langle v, X_i \rangle^2 \mathbf{1}_{F_a}.$$

*Proof.* As announced, fix $\tau > 0$ and let $F_a \triangleq \left\{ \frac{1}{|a|} \sum_{i \in a} \langle v, X_i \rangle^2 \leq \tau \right\}$. We have that:

$$\frac{1}{|a|} \mathbf{E} \left[ \sum_{i \in a} \langle v, X_i \rangle^2 - \sum_{i \in a} \langle v, X_i \rangle^2 \mathbf{1}_{F_a} \right]$$

$$= \mathbf{E} \left[ \frac{1}{|a|} \sum_{i \in a} \langle v, X_i \rangle^2 \mathbf{1}_{F_a^c} \right]$$

$$\leq \sqrt{ \mathbf{E} \left[ \frac{1}{|a|} \sum_{i \in a} \langle v, X_i \rangle^2 \right]^2 \mathbf{P}(F_a^c) } \qquad \text{(Cauchy-Schwarz)}$$

$$\leq \mathsf{h} \sqrt{ \mathbf{E} \left[ \frac{1}{|a|} \sum_{i \in a} \langle v, X_i \rangle^2 \right] } \sqrt{\mathbf{P}(F_a^c)} \qquad \text{(Lemma A.1)}$$

$$\leq \mathsf{h} \sqrt{ \mathbf{E} \left[ \frac{1}{|a|} \sum_{i \in a} \langle v, X_i \rangle^2 \right] } \frac{\sqrt{ \mathbf{E} \left[ \frac{1}{|a|} \sum_{i \in a} \langle v, X_i \rangle^2 \right]^2 }}{\tau^2} \qquad \text{(Markov's Inequality)}$$

$$\leq \frac{\mathsf{h}^2}{\tau^2} \mathbf{E} \left[ \frac{1}{|a|} \sum_{i \in a} \langle v, X_i \rangle^2 \right]. \qquad \text{(Lemma A.1)}$$

$$\text{(A.5)}$$

To finish the proof, multiply both sides of (A.5) above by $|a|$ and re-arrange. ∎

We now proceed with the truncation argument. Define the function

$$\phi_\tau(u) \triangleq \begin{cases} u^2 & \text{if } |u| \leq \tau, \\ \tau^2 & \text{if } |u| > \tau. \end{cases}$$

The truncation is combined with an approximation. Namely, notice that we have

$$\frac{1}{n} \sum_{i=1}^n \langle v, X_i \rangle^2 \geq \frac{1}{n} \sum_{i=1}^n \phi_\tau(\langle v, X_i \rangle)$$

$$= \frac{1}{n} \mathbf{E} \sum_{i=1}^n \phi_\tau(\langle v, X_i \rangle) - \frac{1}{n} \left[ \mathbf{E} \sum_{i=1}^n \phi_\tau(\langle v, X_i \rangle) - \sum_{i=1}^n \phi_\tau(\langle v, X_i \rangle) \right]$$

$$\geq \mathbf{E} \sum_{i=1}^n \phi_\tau(\langle v, X_i \rangle) - \sup_{v \in \partial \Sigma_X} \frac{1}{n} \left[ \mathbf{E} \sum_{i \in O} \phi_\tau(\langle v, X_i \rangle) - \sum_{i \in O} \phi_\tau(\langle v, X_i \rangle) \right]$$

$$- \sup_{v \in \partial \Sigma_X} \frac{1}{n} \left[ \mathbf{E} \sum_{i \in E} \phi_\tau(\langle v, X_i \rangle) - \sum_{i \in E} \phi_\tau(\langle v, X_i \rangle) \right]$$

$$\text{(A.6)}$$

where the last step uses the assumption that $v \in \partial \Sigma_X$.

By invoking Lemma A.2 and noticing that $\phi_\tau$ can be lower-bounded via the indicators of the $F_a$ we have that:

$$\frac{1}{n} \sum_{i=1}^n \mathbf{E} \phi_\tau(\langle v, X_i \rangle) \geq \frac{1}{n} \sum_{i=1}^n \mathbf{E} \langle v, X_i \rangle^2 \mathbf{1}_{F_a} \geq \left(1 - \frac{\mathsf{h}^2}{\tau^2}\right) \left[ \frac{1}{n} \sum_{i=1}^n \mathbf{E} \langle v, X_i \rangle^2 \right] = \left(1 - \frac{\mathsf{h}^2}{\tau^2}\right)$$

$$\text{(A.7)}$$

since $v \in \partial \Sigma_X$. Consequently by combining (A.6) and (A.7) we obtain the deterministic decomposition:

$$\frac{1}{n} \sum_{i=1}^n \langle v, X_i \rangle^2 \geq \left(1 - \frac{\mathsf{h}^2}{\tau^2}\right) - \sup_{v \in \partial \Sigma_X} \frac{1}{n} \left[ \mathbf{E} \sum_{i \in O} \phi_\tau(\langle v, X_i \rangle) - \sum_{i \in O} \phi_\tau(\langle v, X_i \rangle) \right]$$

$$- \sup_{v \in \partial \Sigma_X} \frac{1}{n} \left[ \mathbf{E} \sum_{i \in E} \phi_\tau(\langle v, X_i \rangle) - \sum_{i \in E} \phi_\tau(\langle v, X_i \rangle) \right] \quad \text{(A.8)}$$

for every $v \in \partial\Sigma_X$.

The decomposition (A.8) sets the stage for the last step of the proof: we will apply the block decoupling result of Proposition 4.1 to replace the empirical processes appearing on the right of (A.8) and pay an additional failure probability of $\sum_{i=2}^{2m-1} \beta_X(a_i)$.

Consequently, it only remains to control the supremum of the following truncated and decoupled empirical process (sgn $\in \{O, E\}$):

$$\sup_{v \in \partial\Sigma_X} \frac{1}{n} \left[ \mathbf{E} \sum_{i \in \text{sgn}} \phi_\tau(\langle v, \tilde{X}_i\rangle) - \sum_{i \in \text{sgn}} \phi_\tau(\langle v, \tilde{X}_i\rangle) \right]. \tag{A.9}$$

### A.2.2 Approximation step: bounding (A.9)

We now intend to invoke Talagrand's inequality for independent but not necessarily equally distributed variables [see Klein and Rio, 2005, Theorem 1.1]. To do so, we first need to control the expectation and then the weak variance of the process (A.9). For future reference, we also point out that by construction:

$$\left| \sum_{i \in a} \mathbf{E}\phi_\tau(\langle v, \tilde{X}_i\rangle) - \phi_\tau(\langle v, \tilde{X}_i\rangle) \right| \leq 2\tau^2 |a|. \tag{A.10}$$

We now turn to controlling the expectation of (A.9). Using that the odd (resp. even) blocks of the decoupled process are independent, a straightforward symmetrization argument yields [see e.g. Wainwright, 2019, Proposition 4.11]:

$$\mathbf{E} \sup_{v \in \partial\Sigma_X} \frac{1}{n} \left[ \mathbf{E} \sum_{i \in \text{sgn}} \phi_\tau(\langle v, \tilde{X}_i\rangle) - \sum_{i \in \text{sgn}} \phi_\tau(\langle v, \tilde{X}_i\rangle) \right] \leq \frac{2}{n} \mathbf{E} \left[ \sup_{v \in \partial\Sigma_X} \sum_{j \in [2m], a_j \subset \text{sgn}} \eta_j \sum_{i \in a_j} \phi_\tau\left( \langle v, \tilde{X}_i\rangle \right) \right] \tag{A.11}$$

for a sequence $\eta_j, j \in [2m]$ of Rademacher random variables. Since the map $(\mathbb{R}^d, \|\cdot\|_2) \ni x \mapsto \sum_{j=1}^d \phi_\tau(x_j) \in (\mathbb{R}, |\cdot|)$ is $2\sqrt{d}\tau$-Lipschitz, the vector version of the Ledoux-Talagrand contraction principle now yields [see Maurer, 2016, Theorem 2]:

$$2\mathbf{E} \left[ \sup_{v \in \partial\Sigma_X} \sum_{j \in [2m], a_j \subset \text{sgn}} \eta_j \sum_{i \in a_j} \phi_\tau\left( \langle v, \tilde{X}_i\rangle \right) \right] \leq 4\tau \sqrt{2 \max_{j \in [2m]} |a_j|} \mathbf{E} \left[ \sup_{v \in \partial\Sigma_X} \sum_{i \in \text{sgn}} \eta_i' \langle v, \tilde{X}_i\rangle \right] \tag{A.12}$$

for another sequence $\eta_i', i \in [n]$ of Rademacher random variables.

Obviously,

$$\mathbf{E} \left[ \sup_{v \in \partial\Sigma_X} \sum_{i \in \text{sgn}} \eta_i \langle v, \tilde{X}_i\rangle \right] = \mathbf{E} \left\| \sum_{i \in \text{sgn}} \eta_i \Sigma_X^{-1/2} \tilde{X}_i \right\|_2 \leq \sqrt{\mathbf{E} \left\| \sum_{i \in \text{sgn}} \Sigma_X^{-1/2} \tilde{X}_i \right\|_2^2} \leq \sqrt{nd_X}. \tag{A.13}$$

Combining (A.11),(A.12) and (A.13) we find that:

$$\mathbf{E} \sup_{v \in \partial\Sigma_X} \frac{1}{n} \left[ \mathbf{E} \sum_{i \in \text{sgn}} \phi_\tau(\langle v, \tilde{X}_i\rangle) - \sum_{i \in \text{sgn}} \phi_\tau(\langle v, \tilde{X}_i\rangle) \right] \leq 4\tau \sqrt{2nd_X \max_{j \in [2m]} |a_j|} \tag{A.14}$$

As announced, we also need an upper bound on the variance of the truncated empirical process (A.9):

$$\sup_{v\in\partial\Sigma_X}\mathbf{Var}\left[\mathbf{E}\sum_{i\in\mathrm{sgn}}\phi_\tau(\langle v,\tilde{X}_i\rangle)-\sum_{i\in\mathrm{sgn}}\phi_\tau(\langle v,\tilde{X}_i\rangle)\right]$$

$$=\sup_{v\in\partial\Sigma_X}\sum_{j\in[2m],a_j\subset\mathrm{sgn}}\mathbf{Var}\left[\mathbf{E}\sum_{i\in a_j}\phi_\tau(\langle v,\tilde{X}_i\rangle)-\sum_{i\in a}\phi_\tau(\langle v,\tilde{X}_i\rangle)\right]\quad\text{(Blockwise Indep.)}$$

$$\leq\sup_{v\in\partial\Sigma_X}\sum_{j\in[2m],a_j\subset\mathrm{sgn}}\mathbf{E}\left[\sum_{i\in a_j}\phi_\tau(\langle v,\tilde{X}_i\rangle)\right]^2\quad\text{(Jensen's Ineq.)}$$

$$\leq\sup_{v\in\partial\Sigma_X}\sum_{j\in[2m],a_j\subset\mathrm{sgn}}\tau^2|a_j|\sum_{i\in a_j}\mathbf{E}\langle v,\tilde{X}_i\rangle^2\quad(|\phi_\tau(\cdot)|\leq\tau^2)$$

$$\leq\tau^2\max_{a\subset\mathrm{sgn}}|a|\sup_{v\in\partial\Sigma_X}\sum_{i\in\mathrm{sgn}}\mathbf{E}\langle v,\tilde{X}_i\rangle^2$$

$$\leq\tau^2 n\max_{j\in[2m]}|a_j|.$$

(A.15)

Combining Klein and Rio's version of Talagrand's inequality with our estimates (A.10), (A.14) and (A.15), we find for any $u\in[0,\infty)$ that:

$$\mathbf{P}\left(\sup_{v\in\partial\Sigma_X}\left[\sum_{i\in\mathrm{sgn}}\left(\mathbf{E}\phi_\tau(\langle v,\tilde{X}_i\rangle)-\phi_\tau(\langle v,\tilde{X}_i\rangle)\right)\right]\geq 4\tau\sqrt{2nd_X\max_{a\subset\mathrm{sgn}}|a|}+u\right)$$

$$\leq\exp\left(\frac{-u^2}{2\mathsf{v}+6\tau^2\max_{j\in[2m]}|a_j|u}\right)\quad\text{(A.16)}$$

with $\mathsf{v}=\tau^2 n\max_{j\in[2m]}|a_j|+8\sqrt{2nd_X\max_{a\subset\mathrm{sgn}}|a|}$.

### A.2.3   Finishing the proof of Theorem 4.3

Fix a positive constant $\alpha\in\mathbb{R}$ to be determined later. We now set $u=\alpha n$. Let us further fix a positive constant $\varepsilon\in\mathbb{R}$ and assume that $n\geq(1/\varepsilon^2)d_X\max_{j\in[2m]}|a_j|$. Under these additional hypotheses:

$$4\tau\sqrt{2nd_X\max_{j\in[2m]}|a_j|}\leq 4\tau n\sqrt{2}\varepsilon\quad\text{(A.17)}$$

and

$$2\mathsf{v}+6\tau^2\max_{j\in[2m]}|a_j|u=2\tau^2 n\max_{j\in[2m]}|a_j|+16\sqrt{2nd_X\max_{j\in[2m]}|a_j|}+6\tau^2\max_{j\in[2m]}|a_j|u$$

$$\leq\tau^2\max_{j\in[2m]}|a_j|(2n+6\alpha n)+16\tau n\sqrt{2}\varepsilon\quad\text{(A.18)}$$

We now decouple using Proposition 4.1 and instantiate our upper bound (A.16) to control (A.8). This step combined with the estimates (A.17) and (A.18) yields:

$$\mathbf{P}\left(\forall v\in\partial\Sigma_X\ :\ \frac{1}{n}\sum_{i=1}^n\langle v,X_i\rangle^2\geq\left(1-\frac{\mathsf{h}^2}{\tau^2}\right)-8\tau\sqrt{2}\varepsilon-2\alpha\right)$$

$$\geq 1-2\exp\left(-\frac{\alpha^2 n}{\tau^2\max_{j\in[2m]}|a_j|(2+6\alpha)+16\tau\sqrt{2}\varepsilon}\right)-\sum_{i=2}^{2m-1}\beta_X(a_i).\quad\text{(A.19)}$$

By choosing $\alpha$ and $\varepsilon$ sufficiently small and $\tau$ sufficiently large, we find that there exists a universal positive constant $C\in\mathbb{R}$ such that if

$$n\geq C\max_{j\in[2m]}|a_j|(d_X+\mathsf{h}^2\log(1/\delta))\quad\text{and}\quad\sum_{i=2}^{2m-1}\beta_X(a_i)\leq\frac{\delta}{2}$$

then

$$\mathbf{P}\left(\forall v \in \partial\Sigma_X \ : \ \frac{1}{n}\sum_{i=1}^n \langle v, X_i\rangle^2 \geq \frac{1}{2}\right) \geq 1-\delta.$$

The result follows by rescaling to arbitrary $v \in \mathbb{R}^{d_X}$. ∎

### A.3 Finishing the of Theorem 3.1

The proof of Theorem 3.1 now easily follows from Theorem 4.3 and Theorem 4.2. We translate the necessary conditions below.

First, we note that in light of Theorem 4.3 if (4.7) holds then we may estimate $\tilde{\Sigma}_n^{-1} \preceq cI_{d_X}$ for some universal positive constant $c > 0$. We remark that (4.7) holds by (3.3) and (3.5). Consequently, it suffices to control the random walk (2.6).

Second, we turn to the control of the random walk (2.6), which is provided by Theorem 4.2, by instantiating it to the case of the Euclidean space $(\mathbb{R}^{d_Y \times d_X}, \|\cdot\|_F)$. We identify this space by linear isometry (vec) with $(\mathbb{R}^{d_Y d_X}, \|\cdot\|_2)$ and hence in the notation of Theorem 4.2 we have that the weak variances are:

$$\Lambda_O = \frac{1}{|O|}\left\|\sum_{i\in[2m], i\in 2\mathbb{N}-1}\mathbf{E}\left[\mathsf{vec}\left(\sum_{j\in a_i}Z_j\right)\mathsf{vec}\left(\sum_{j\in a_i}Z_j\right)^{\mathsf{T}}\right]\right\|_{\mathsf{op}};$$

$$\Lambda_E = \frac{1}{|E|}\left\|\sum_{i\in[2m], i\in 2\mathbb{N}}\mathbf{E}\left[\mathsf{vec}\left(\sum_{j\in a_i}Z_j\right)\mathsf{vec}\left(\sum_{j\in a_i}Z_j\right)^{\mathsf{T}}\right]\right\|_{\mathsf{op}}.$$

Using monotonicity of the operator norm over symmetric positive semidefite matrices and the condition (3.4) we also have for either $\mathsf{sgn} \in \{O, E\}$:

$$c'\sigma^2 \leq \Lambda_{\mathsf{sgn}} \leq \sigma^2 \tag{A.20}$$

for some universal positive constant $c'$. Condition (3.4) also implies that the constant $r$ in Theorem 4.2 can be chosen as $c''\mathsf{edim}(\Sigma)$ for some universal positive constant $c''$. The final step is to note that the last two terms on the right hand side of (4.6) are less than $c'''\delta$ for some third universal positive constant $c'''$ if the respective second parts of (3.3) and (3.4) hold. ∎

