# OpenReview forum: "The noise level in linear regression with dependent data"
_NeurIPS.cc/2023/Conference — NeurIPS 2023 poster_

### Official Review · Reviewer_6Vi5 · 2023-06-23

**Soundness:** 4 excellent
**Presentation:** 4 excellent
**Contribution:** 3 good
**Rating:** 7
**Confidence:** 4

**Summary:**

The focal point of this paper is very precise, namely least-squares linear regression under data which need not be independent. Regardless of linearity, when the model is properly specified (realizability, i.e., the expected squared error minimizer is included in the model), martingale-based arguments are well-established in the literature. It is this realizability assumption that the authors remove. They conduct a blocking-based argument; this breaks up the data into blocks which are essentially independent, but takes a hit because the effective sample size is reduced. Their blocking-based argument is done in a careful way, such that the resulting "noise level" factor (a variance-like quantity) that appears in excess risk bounds is not excessively inflated by this reduction.

**Strengths:**

The paper is extremely well-written, with notation and exposition all crystal-clear. The main content is all quite technical in nature, but the key ideas are explained in an intuitive fashion, with appropriate references to the relevant literature upon which this work stands. The paper is centered around the main technical result (Thm 3.1), and is organized to give relevant background to understand the technical context in which this result stands, and to describe the essential points of their proof. In my opinion, the balance between informal and formal content is excellent.

The main result of this work is Theorem 3.1. While there are several technical "burn-in" conditions (the sample cannot be too small, relative to the degree of dependence and noise level, etc.), the core result (3.2) is clear, appealing, and to the best of my knowledge fills a valid gap in the theoretical literature for linear least-squares with dependent data.

**Weaknesses:**

Obviously this is a technical paper with a very particular problem of interest, and thus the overall "impact" on the field of machine learning is limited, but the solution provided by the authors to this problem is presented clearly, and the main claims are to the best of my understanding solid.

The only point I had trouble with in terms of clarity was the notion of "instance-specific" and "instance-optimal" performance guarantees. I know the authors try to spell this out in the first paragraphs of 3.1, but if space allows, I think a more explicit explanation of the "global" complexity in previous works would make the "local" complexity here a lot more clear.

A couple other small points:
- Since there is some space left, I felt that it would be nice to give $\\widehat{M}$ a definition analogous to $M\_{\\star}$ in (2.1), instead of just giving the form in (2.4) and saying it is the OLS solution. I know this is simple to verify, but if space allows it, such an explicit expression makes it more friendly to a crowd familiar with notions of ERM.
- The last sentence of paragraph 2 in section 1 is repeated.

**Questions:**

I do not have any points to confirm that would change my opinion of the paper.

**Limitations:**

The technical assumptions are all given explicitly, with plentiful references to the existing literature, so I feel the limitations of this work are quite clear.

---

> ### Author Rebuttal · Authors · 2023-08-03
>
> We thank the reviewer for their time and effort spent on evaluating our manuscript.
>
> * We have clarified the distinction between local and global complexities by adding the following sentence to Section 3.1: "In other words, they compete against the worst distribution at a given level of mixing, whereas we compete against a fixed  distribution."
>
> * We have also added a short remark that $\widehat M$ is an empirical risk minimizer at the end of the problem formulation as per your request.

---

> ### Comment · Reviewer_6Vi5 · 2023-08-18
> **Re: Rebuttal by Authors**
>
> I thank the authors for their response. Having read the other reviews, my overall opinion remains the same.

---

### Official Review · Reviewer_2Nqt · 2023-07-03

**Soundness:** 3 good
**Presentation:** 3 good
**Contribution:** 2 fair
**Rating:** 4
**Confidence:** 4

**Summary:**

This paper deals with linear regression with dependent ($\beta$-mixing) data. It provides an upper bound of the OLS error in terms of the sample size and the effective dimension of the covariate matrix.

**Strengths:**

This paper studies the linear regression with dependent data. The main idea is to decompose the data into blocks, and apply the concentration inequalities. The authors also provide bounds on the burn-in period. The overall results are new, though the idea does not seem to be novel. The presentation of the paper is clear, and I mostly enjoyed reading the paper.

**Weaknesses:**

The main weakness of this paper is that the approach of combining blocking with concentration is quite traditional. As pointed out by the authors themselves, this idea was carried out in [10] (almost 30 years ago), with subsequent development by Massart, Wu...etc. The main result (Theorem 3.1) is more or less expected in the high dimensional statistics. Moreover, there is no (empirical) experiment for illustration.

**Questions:**

I have no specific question, but one general comment. The result established in this paper has close affinity to nonparametric estimation (with possibly non i.i.d. noise, e.g. time series). This is a vast field, and the authors may want to add more discussions on this connection, see e.g. Wu's paper Nonlinear system theory: another look at dependence. PNAS, 2005.

---

> ### Author Rebuttal · Authors · 2023-08-03
>
> We thank the reviewer for their time and effort reviewing our submission.
>
> By our estimation, the reviewer's main concern is a lack of novelty, in the sense that similar estimates on the  random walk component of our analysis (i.e. the "numerator" in the estimation error)  have appeared previously in the large/moderate deviations literature.  We do not argue this fact but would like to  some nuance to claim that the result is "expected" or lacks novelty and kindly ask the reviewer to reconsider their score in light of this if possible.
>
> First, one of the main points of our manuscript is precisely that this is expected (from the central limit theorem) but that, nevertheless, sharp estimates for learning with dependent data are conspicuously absent in the literature (beyond certain special cases as we note).  Indeed, a quick glance at our references (and more can probably be found) reveals that several top tier venues including NeurIPS, the Annals of Statistics, and COLT have published results on square loss  quite recently in which the noise term is deflated by such mixing time dependencies. Moreover, just as we do here, some of these papers focus entirely on linear classes (e.g. Wong et al. (Ann.Stat. 2020) and Nagaraj et al. (NeurIPS 2020)).  Hence, even if these papers combine blocking with concentration, they do not arrive at rates that match known asymptotics---again: remedying this is precisely the point of the present manuscript.
>
> Arguably, we then have that 1) the problem is of interest to the intended audience; 2)  to the best of our knowledge, there are no previous works that manage to obtain sharp non-asymptotics for any convex class with general dependent data---even though they as we do combine blocking and concentration, they only manage to obtain multiplicative instead of additive dependency on mixing; and 3) miss-specified linear regression is a natural starting point for sharpening this dependency.
>
> With this in mind then, it appears a little unfair to us to reject our solution based on the fact that it is "simple" and draws on existing ideas from probability theory.

---

> > ### Comment · Reviewer_2Nqt · 2023-08-11
> >
> > Many thanks for the detailed explanations. The score remains unchanged.

---

### Official Review · Reviewer_8guK · 2023-07-10

**Soundness:** 2 fair
**Presentation:** 2 fair
**Contribution:** 1 poor
**Rating:** 3
**Confidence:** 4

**Summary:**

The paper explores the impact of noise level in linear regression for dependent data by blocking technique, which can accommodate a broad type of dependent structures.  Theoretical justification of the non-asymptotic guarantee and excess risk bound  are provided, imposing any realizable assumptions on the noise.  The paper's insights and conclusions are likely to influence and inspire further research in the field.


**Strengths:**

The authors propose a novel perspective by combining  $\beta$-mixing assumption and blocking technique, leading to a new approach for addressing the dependent data. This innovative combination offers unique insights and contributes to advancing the field in handling such data structures. This paper demonstrates a solid theoretical foundation, characterized by sound reasoning and logical coherence. The non-asymptotic results are precisely analyzed, and potential limitations are appropriately addressed.


**Weaknesses:**

1. Inadequate comparison with prior work: The paper could benefit from a more thorough comparison with existing literature. Providing a detailed analysis and critique of related works would help situate the proposed approach in the context of prior research. Identifying the limitations of previous methods and explicitly explaining how the proposed approach addresses those limitations would strengthen the argument for the paper's contribution.
2. Lack of clarity in methodology: The paper could benefit from providing more detailed and explicit explanations of the research introduction and methodology. Some sections may be unclear or lack sufficient technical details, making it difficult for readers to understand the implementation nuances. Adding supplementary information, such as mathematical formulations, pseudo-code, or algorithmic details, would greatly improve the clarity of the work.
3. Lack of experimental validation: There conducted no numerical results in the manuscript, which are limited in scope and fail to provide a comprehensive evaluation of the proposed approach. To strengthen the research, the authors should consider adding some numerical experiments  to demonstrate the effectiveness and generalization of their method compared to existing methods. Additionally, providing a comparative analysis with existing methods would further highlight the strengths and weaknesses of the proposed method.

**Questions:**

There are some issues need to be addressed.
1. The martingale technique is commonly-used to address the dependence in random stochastic process. Could the authors highlight the differences between the proposed methods and martingale? Further, it would be valuable for the authors  to give some discussion about the realizable and non-realizable learning under dependent setting.

2. In the problem formulation, does the last equality hold in the definition of excess risk in Equation (2.3)?

3. The authors claimed that the sidestep of blocking can be decomposed as controlling the lower tail of the empirical covariance matrix and the interaction between the noise and covariates,  and thus adopt $\beta$-mixing assumption to address the interaction term for non-asymptotic bound.  However, I wonder what is the main contribution of the proposed method except for the use of  $\beta$-mixing and blocking, since the two instruments are very common to tackle dependent data or time series. Maybe the Summary section need to be rewritten.

4.  In Theorem 3.1, it would be better if the detail of correcting the blocking inflation factor is exposed.

5. The authors mentioned that the non-asymptotic result allows for arbitrary dimensions and unbounded process, but there is no further discussion or elaboration on this aspect.

6. The authors have mentioned  that the proposed method outperform the some existing work. It would be beneficial if the authors could conduct some numerical experiments to support the proved theoretical results compared to some existing works, for example, Nagaraj et al. (2020).

7. The manuscript is not well-organized and there are many typos in the manuscript and the writing  needs more polishing. For example, in Section 1 of Page 1 Paragraph 2, the last sentence appeared twice.


**Limitations:**

No other major concerns than the ones listed in the weaknesses and questions.

---

> ### Author Rebuttal · Authors · 2023-08-03
>
> We do not agree with the assessment provided by reviewer 8guK and believe it should be disregarded. There is no real criticism of our work in this review other than sweeping and unsubstantiated (and sometimes contradictory) claims. The main points of contention appear to be 1) related work, 2) clarity, and 3) experiments. We address these below.
>
> 1. While we do not claim to be exhaustive, we believe we have provided a fair overview of closely related contributions. In particular, we have provided a detailed comparison to the work of Nagaraj et al. in Section 3.1, which to the best of our knowledge is most closely related.
> We also note that the other reviewers appear to be in agreement with us on this point.
>
> 2. The reviewer believes that our work "lacks clarity in methodology". We find this criticism to be rather sweeping and without proper justification. We also note that the other reviewers appear to find the paper rather clear.
>
> 3. While experiments generally do not hurt we do not think they would do much to improve this particular paper, and in particular we do not agree with the way this criticism is written by the reviewer. Our result is a refined analysis of a very standard algorithm and in particular we do not provide any new algorithm and so providing a "comparative analysis with existing methods" does not make much sense.
>
> We also note that the review is somewhat incoherent, examples including for instance that the paper is likely to "inspire further research in the field" while being awarded a contribution score of "1".
>
>
>
> We respond below to the reviewer's questions below:
>
> 1. As is stated in the introduction, the differences lie in the set of assumptions. This in turn necessitates a new proof approach as is provided here.
> 2. Yes, this is a well-known identity for the population risk in the regression literature and is easily obtained invoking optimality of $M_\star$ to the population risk criterion.
> 3. As is clearly stated at several places throughout the paper, the main contribution is that we are first to provide a sharp non-asymptotic analysis of least squares for a wide range of processess ($\beta$-mixing). Previous results operate either under  strict realizability assumptions or deflate the rate by a "mixing time factor". With regard to the use of blocking and mixing the novelty lies in the way these are combined.
> 4. We could not follow the meaning of this question. Could the reviewer clarify the meaning of exposing the blocking inflation factor? We have clearly stated our burn-in conditions.
> 5. As the reviewer notes we state in the introduction that the one dimensional "Bernstein sketch" can be extended to arbitrary dimensions and unbounded processes. This is precisely the statement of Theorem 3.1 which holds for covariates taking values in any finite dimensional space and does not impose boundedness assumptions on these covariates other than the existence of 4 moments. We are left wondering what the reviewer means?
> 6. We believe the rather detailed comparison in Section 3.1 makes this point sufficiently clear.
> 7. We thank the reviewer for point out the accidental doubling of this sentence. We do however strongly disagree with the suggestion that the manuscript is not well-organized and it appears strange to us in light of how highly the other reviewers rate our presentation.

---

> > ### Comment · Reviewer_8guK · 2023-08-21
> >
> > Thank you for your reply. I've gone through the authors' rebuttal, but my score will remain the same.

---

### Official Review · Reviewer_vLcM · 2023-07-21

**Soundness:** 3 good
**Presentation:** 4 excellent
**Contribution:** 3 good
**Rating:** 6
**Confidence:** 3

**Summary:**

This paper studies the risk bounds of OLS for linear regression with dependent data. In particular, the label noise is allowed to be non-martingale. It shows that, after a burn-in phase, OLS with dependent data archives a bound of the same order as if the data is iid, provided that the failure probability is moderately small. The proved bound is particularly interesting because it suggests that the leading error does not explicitly depend on the mixing time.

**Strengths:**

+ Excellent presentation.

+  Sharp risk bounds are well-developed for OLS for linear regression with iid data. However, when the data is dependent, the risk achieved by OLS is less clear. Surprisingly, this work proves that, even when the data is dependent, the risk of OLS still recovers that predicted by the central limit theorem and does not rely on the mixing time, given that the sample size exceeds a burn-in requirement and that the failure probability is moderately small.

+ The proof is decomposed into two neat parts, the first part controls a dependent random walk with the blocking technique and the other part controls the lower tail of the empirical covariance. The proof demonstrates the new ingredients of this work that allows obtaining the improved bound.

+ Prior works are well discussed. I especially appreciate the comparison with [23], which clarifies that, in the worst case, the bound will depend on the mixing time, but in average cases, the bound does not explicitly depend on the mixing time.





**Weaknesses:**

Please see the questions below.

**Questions:**

1. My main question is regarding line 158, the "hypercontractivity condition". To my knowledge, in the iid case, the hypercontractility condition states that
$$ E \langle v, x \rangle^4 \le h^2 \cdot \langle v, E [xx^\top] v  \rangle^2. $$
In comparison, the "hypercontractivity condition" introduced in line 158 misses a square on the right-hand side. Could you please comment on this difference? I understand that the "hypercontractivity condition" introduced in line 158 only requires $v$ such that
$$ \langle v, \Sigma v\rangle = 1.$$
Will the "hypercontractivity condition" in line 158 implies an additional condition on the data dependence?

2. Line 182. ```The first burn-in condition of (3.3) is standard for control of the lower tail... it is optimal even in the iid regime.```
I am not sure if this statement is accurate. For example, the work by [BLLT] (and its many follow-ups) allows $n$ to be smaller than $d_{X}$. Granted, they do need stronger assumptions such as data being iid and sub-Gaussian.

3. One drawback of the bound is that the fail probability $\delta$ cannot be made arbitrarily small. The condition (3.5) implies the mixing time has to be sufficiently fast. This is also acknowledged in lines 192-197.

4. Line 169. ```The scaling with ... thus scales as expected in the iid regime, but also degrades gracefully with dependence. We reiterate that (3.2) does not depend directly on mixing in the leading order term....```.
I feel the discussion is a little confusing. Note that even if the bound has a multiplicative dependence on the mixing time, it still degrades "gracefully" with dependence. Because in the iid case, the mixing time is just $1$ so a multiplicative dependence does not harm either.


## Typos
1. Line 52. Are there some typos in the definitions of $\bar V_i$? I guess the right definition be
$$ \bar V_i = \sum_{j = (i-1)k + 1}^{ik} V_j$$

2. Line 128. "empirical excess risk" -> "excess risk"

3. Should mention that eq (3.2) holds with probability $> 1-\delta$.



[BLLT] Bartlett, Peter L., Philip M. Long, Gábor Lugosi, and Alexander Tsigler. "Benign overfitting in linear regression." Proceedings of the National Academy of Sciences 117, no. 48 (2020): 30063-30070.

---

> ### Author Rebuttal · Authors · 2023-08-03
>
> We thank the reviewer for their time and effort spent on reviewing our manuscript. We provide brief answers to their questions 1,2 and 4 below (with 3 being more of a statement).
>
> * Answers to Questions.
>
> 1. (hypercontractivity/moment equivalence). For sake of argument, let us assume that the process in question is stationary. The condition we use is then equivalent to $L^4/L^2$ equivalence on the $\Sigma_X$-sphere. Namely for fixed $v\in \partial\Sigma_X$, the requirement that $\mathbb{E} \langle v,X\rangle^4 \leq \langle v, \mathbb{E} XX^\top v\rangle^2$ is exactly the same as the requirement that  $\mathbb{E} \langle v,X\rangle^4 \leq \langle v, \mathbb{E} XX^\top v\rangle$, precisely because  $\langle v, \mathbb{E} XX^\top v\rangle=1$ since $\partial\Sigma_X = \{ v : \langle v, \mathbb{E} XX^\top v\rangle=1 \}$ by definition.  Put differently, these conditions only differ significantly if one requires them to hold outside the $L^2_{P_X}$-unit-sphere but since we are working with a linear class we only need to consider the unit ball (by a rescaling argument). In general (without stationarity) these conditions will not differ by more than a factor of a condition number between $\Sigma_X$ and $\Sigma_{X_i}$.
>
> 2. This is a good point, thank you. We should have phrased this more carefully so that it refers to invertibility of the empirical covariance matrix---changes to reflect this have been made.
>
> 3. This is correct---these burn-ins are obtained  by requiring that various additive terms are "sufficiently small". We could in principle present a bound valid without these mixing-related burn-ins but there is no guarantee that our results are sharper than existing bounds for $\delta$ in this regime.
>
> 4. We will clarify this point, thank you.
>
> Typos: We thank the reviewer for pointing these out to us.

---

> > ### Comment · Reviewer_vLcM · 2023-08-10
> > **Regarding hypercontractivity**
> >
> > I understand that your condition is only stated for vectors in a unit ellipse so it is equivalent to the conventional version in the stationary case. My concern was that
> > ```
> > Will the "hypercontractivity condition" in line 158 implies an additional condition on the data dependence?
> > ```
> > In the response, you have suggested that
> > "In general (without stationarity) these conditions will not differ by more than a factor of a condition number between $\Sigma_X$ and $\Sigma_{X_i}$"
> >
> > I believe that missing a condition number between $\Sigma_X$ and $\Sigma_{X_i}$ is a significant caveat and should have been clarified in the paper. Also, it would be helpful to give examples to demonstrate when this condition number is small and how this requirement affects the data dependence.

---

> > > ### Author Response · Authors · 2023-08-11
> > >
> > > Many thanks for taking the time to clarify your question.
> > >
> > > First, just to be clear, let us state that any condition number that does appear will be relegated into the factor $\mathsf{h}$ which only appears in the burn-in. It does not affect the final rate in the leading order term.
> > >
> > > Now, you are correct in stating that the some degree of well-conditioning is necessary. Although we believe this has more to do with us requiring recovery of $M_\star$ in $\Sigma_X$-norm (as opposed to some weaker 2-norm given by $\Sigma' \prec \Sigma_X$) than dependence directly.
> > >
> > > With regards to the effect on imposing dependence structure, it is instructive to first consider what happens for $\mathsf{X}$ a compact state space. In this case, $X_i X_i^\top$ is uniformly bounded and the condition we impose holds trivially with $\mathsf{h}$ exhibiting dependence on the diameter of $\mathsf{X}$ and the smallest eigenvalue of $\Sigma_X$. Hence, at least for bounded covariates no further assumption on dependence is necessary to verify our condition.
> > >
> > > Now, beyond boundedness the question is actually rather subtle even if we do not impose any dependence structure. Namely, one can construct regression tasks with independent but _not_ identitcally distributed covariates in which an exponential (in dimension) blow-up in the risk is unavoidable unless the covariances are well-conditioned in some sense. A detailed statement can found as Theorem 6.2 in [1].
> > >
> > > We will clarify this following the main Theorem statement and are of course also happy to include further discussion to flesh out these subtleties in an updated version of the manuscript.
> > >
> > > [1] Tu, Stephen, Roy Frostig, and Mahdi Soltanolkotabi. "Learning from many trajectories." arXiv preprint arXiv:2203.17193 (2022).

---

### Official Review · Reviewer_BbWm · 2023-07-25

**Soundness:** 3 good
**Presentation:** 3 good
**Contribution:** 3 good
**Rating:** 7
**Confidence:** 3

**Summary:**

The paper gives finite-sample bounds on the excess risk of ordinary least squares regression in the non-realizable case with dependent ($\beta$-mixing) data. The result asymptotically matches the predictions of the central limit theorem. The dependence on the mixing behaviour of the process is relegated to terms of smaller order in $1/{n}$, where $n$ is the length of the observed sample path, so that, apart from absolute constants, the bound asymptotically coincides with those for independent data.

The technical proof decomposes the excess risk as the norm of the product of a weighted random walk and the inverse covariance matrix (prefiltered with the true covariance). These two factors are bounded separately, in both cases using the blocking technique, which has become a standard method when dealing with $\beta$-mixing processes. How it is avoided, that the mixing times enter the  asymptotically dominant term of the bound, is already explained in the introduction by the example of Bernstein's inequality. The technical details are a major achievement (I did not have time to verify all of it) and relegated to the appendix.

**Strengths:**

The paper addresses an important and obvious problem and offers a largely satisfactory solution.

The multiplicative dependence on the mixing times is a major problem which besets many bounds for dependent data. It is a major accomplishment to free the dominant term of the bound from this dependence. The illustrative explanation in section 1.1 (lines 44-57) is very nice.

I did not find any faults, but because of time constraints I could not verify all the material in the appendix. Otherwise I would have rated the soundness with 4 rather than 3.


**Weaknesses:**

The statement of Theorem 3.1 is somewhat opaque because of the choice of the blocking partition. Presumably the cardinalities of the partition members have to be different to accommodate the non-stationarity of the process, as specified in eq (3.4).
The theorem would be more transparent if first stated for stationary processes using a homogeneous partition. The more general version could be stated in the supplement.

The most critical "burn-in" condition seems to be the second condition in 3.3, because of its dependence on $\sigma^2$ and $\delta$ and in particular if $s=4$. How shall we interpret the slow-down of the "burn-in" as $\sigma^2$ becomes small? It seems to me that this condition would merit a more detailed discussion.

**Questions:**

Suggestions:

Thm 3.1 first stated for stationary processes.

Detailed discussion of the second condition in (3.3)

**Limitations:**

The limitations seem to be adequately addressed.

---

> ### Author Rebuttal · Authors · 2023-08-03
>
> We thank the reviewer for their time and effort spent on evaluating our manuscript.
>
>
> * While we agree that there is trade-off between generality, obtuseness and being concise, we prefer to leave the statement of Theorem 3.1 as is. Our motivation for this is that we already have an informal theorem statement which is relatively easy to parse.
>
>
> * (Relating to the appearance of $\sigma^2$ in the denominator in the second burn-in condition). Note that it is in fact a ratio-squared of the $s$:th moment and the 2nd moment that appears in this condition. This ratio appears because we use higher moments than $2$ to control the deviation of a normalized random walk from its 2nd moment (in principle by Markov's inequality). We will clarify this point in the manuscript.

---

> > ### Comment · Reviewer_BbWm · 2023-08-16
> >
> > Thank you for the clarification of the second point.
> > I find the "informal theorem" a bit too informal. I had hoped for a precise version for the stationary case.

---

### Decision · Program_Chairs · 2023-09-21

**Decision:**

Accept (poster)

**Comment:**

This is a surprising and nontrivial result, on a fundamental topic that has already attracted wide interest.

I strongly encourage the authors to take BbWm's advice and write a more easily understood theorem, possibly in a special case as a corollary.